# Potential Role of Glucagon-like Peptide-1 Receptor Agonists in the Treatment of Cognitive Decline and Dementia in Diabetes Mellitus

**DOI:** 10.3390/ijms241411301

**Published:** 2023-07-11

**Authors:** Maria Chiara Pelle, Isabella Zaffina, Federica Giofrè, Roberta Pujia, Franco Arturi

**Affiliations:** 1Unit of Internal Medicine, Department of Medical and Surgical Sciences, University “Magna Graecia” of Catanzaro, 88100 Catanzaro, Italy; mcpelle@unicz.it (M.C.P.); zaffina86@gmail.com (I.Z.); federica.giofre@gmail.com (F.G.); roberta.puj@gmail.com (R.P.); 2Research Center for the Prevention and Treatment of Metabolic Diseases (CR METDIS), University “Magna Graecia” of Catanzaro, 88100 Catanzaro, Italy

**Keywords:** inflammation, GLP-1, diabetes, dementia, cognitive decline, diabetic complications

## Abstract

Dementia is a permanent illness characterized by mental instability, memory loss, and cognitive decline. Many studies have demonstrated an association between diabetes and cognitive dysfunction that proceeds in three steps, namely, diabetes-associated cognitive decrements, mild cognitive impairment (MCI; both non-amnesic MCI and amnesic MCI), and dementia [both vascular dementia and Alzheimer’s disease (AD)]. Based on this association, this disease has been designated as type 3 diabetes mellitus. The underlying mechanisms comprise insulin resistance, inflammation, lipid abnormalities, oxidative stress, mitochondrial dysfunction, glycated end-products and autophagy. Moreover, insulin and insulin-like growth factor-1 (IGF-1) have been demonstrated to be involved. Insulin in the brain has a neuroprotective role that alters cognitive skills and alteration of insulin signaling determines beta-amyloid (Aβ) accumulation, in turn promoting brain insulin resistance. In this complex mechanism, other triggers include hyperglycemia-induced overproduction of reactive oxygen species (ROS) and inflammatory cytokines, which result in neuroinflammation, suggesting that antidiabetic drugs may be potential treatments to protect against AD. Among these, glucagon-like peptide-1 receptor agonists (GLP-1RAs) are the most attractive antidiabetic drugs due to their actions on synaptic plasticity, cognition and cell survival. The present review summarizes the significant data concerning the underlying pathophysiological and pharmacological mechanisms between diabetes and dementia.

## 1. Introduction

The prevalence of diabetes as well as the longevity and aging of the population are rapidly increasing worldwide [1]. According to the World Health Organization (WHO), 400 million individuals have diabetes, and this number is predicted to double over the next 20 years. Diabetes mellitus (DM) is associated with a high frequency of morbidity and mortality [2]. DM involves several organs, resulting in chronic hyperglycemia-related systemic complications, which can be classified into macrovascular and microvascular injuries. Several studies have demonstrated that DM is associated with additional serious public health problems, namely, cognitive decline and dementia [3,4,5,6]. Numerous studies have shown that patients with type 2 diabetes mellitus (T2DM) [7] have an approximately 1.5–3 times greater risk of developing Alzheimer’s disease (AD) or vascular dementia [8]. Studies have demonstrated a close association between glycated hemoglobin (HbA1c) and the risk to develop dementia, with every 1% of increase in HbA1c correlating with a significant reduction in scores on various cognitive function tests, indicating that chronic hyperglycemia (rather than a solitary glycemic spike) generates the risk [9]. Several hypotheses have been proposed to explain this augmented risk of dementia in the presence of DM. First, hypoglycemia- and hyperglycemia-related microvascular changes can result in microinfarctions [10]. Biessels G. J et al. (2006) reported that the incidence of Alzheimer’s disease and vascular dementia was higher in subjects with diabetes than in those without diabetes [10]. This paper suggested that impaired glucose metabolism and vascular disease underlies the pathophysiology of dementia. In particular, chronic hyperglycemia in diabetes can lead to brain structure abnormalities, such as thickening of the basement membrane of cerebral capillaries and ultimately chronic brain ischemia [11,12]. Similarly, hypoglycemia frequently associated with insulin treatment can cause irreversible structural brain abnormalities [13]. Indeed, several studies demonstrated that the risk of dementia was highest in insulin-treated diabetic people [13]. Director indirect effects of insulin could contribute to the risk of dementia. Hypoglycemic episodes frequently complicate insulin treatment and contrary to hyperglycemia, hypoglycemia can cause irreversible brain damage [14].

Another possible pathophysiological mechanism that may explain this relation is that high levels of HbA1c cause an altered effect of intracerebral insulin signaling pathways due to insufficient secretion, reduced activity, or both [9], resulting in a loss of cellular ionic homeostasis, oxidative stress, an increase in β-amyloid (Aβ) deposits, and phosphorylation of tau proteins [9].

Diabetic cognitive dysfunction is divided into the following three phases: diabetes-associated cognitive decrements, mild cognitive impairment (MCI), and dementia [15]. Hence, in the present review, we discuss the correlations between diabetes and cognitive decline as well as between diabetes and AD, stressing underlying pathophysiological mechanisms of diabetic cognitive disfunction and brain insulin resistance. Further, we discuss the effects of glucagon-like peptide-1 (GLP-1) on neuroinflammation and its possible therapeutic role in both dementia and AD.

## 2. Diabetes, Cognitive Dysfunction and Dementia

Cognitive dysfunction is increasingly recognized as an important comorbidity of T2DM, and epidemiological evidence supports an association between diabetes and cognitive decline. There are important and shared pathological features of T2DM with cognitive decline and dementia, which are characterized by metabolic alterations in the brain, e.g., insulin resistance, altered glucose uptake, and altered glucose utilization. These similarities in pathology are reflected in clinical studies that have demonstrated an increased risk of dementia in individuals with T2DM [9,16,17]. Dementia is an umbrella term, which encompasses several distinct clinical entities. Dementia is a chronic, irreversible condition marked by mental instability, memory loss, and cognitive decline. Cognitive impairment (CI) progresses with time along a continuum, starting from MCI leading to dementia. The most common dementia is AD, which is defined by the presence of extracellular Aβ plaques and intracellular hyperphosphorylated tau tangles in the brain. In addition, vascular dementia occurs in 20 to 30% of cases [16,17]. 

Importantly, individuals with type 1 diabetes mellitus (T1DM) may experience alterations in cognitive development, especially in patients with an early onset of diabetes, but this cognitive decline remains stable over the years [18]. Compared to individuals without diabetes, individuals with T2DM have difficulties in memory processes, executive function, and analysis of unstructured information [19]. So there are differences between T1DM and T2DM: the first one appears to be associated to a slow-down of mental processing and executive/attentional functioning, while the second one has additionally memory problems [20]. Hyperglycemia and microvascular complications are risk factors that we found both in T1DM and T2DM.However, T2DM is diagnosed at an older age than T1DM (old age is in itself a risk factor for dementia) and it occurs with obesity, insulin resistance, hypertension, and dyslipidemia, which have negative effects on the brain [21]. On the other hand, T1DM is almost always diagnosed during childhood and adolescence that are periods of development of central nervous system and younger brain is more susceptible to extreme values of glycemia [22]. Initially, in T2DMthe cognitive alterations may not affect daily life and may not be evident [7]. This cognitive decline results from a reduced ability of brain processing similar to the alteration observed in cognitive aging [15], which starts in the prediabetic phase and proceeds more rapidly than cognitive aging. T2DM-related cognitive dysfunction is divided into the following three stages with an increasing level of severity: (1) diabetes-associated cognitive decrements, (2) MCI (non-amnesic MCI and amnesic MCI), and (3) dementia (both vascular dementia and AD) [15]. The first stage may not affect daily life, and it appears as slight cognitive changes in few fields, such as brain speed and memory [7]. MCI includes declines in some cognitive domains with a subtle impact on daily life [23]. According to prospective population-based studies, diabetes increases the risk to develop amnesic and non-amnesic MCI [24]. Non-amnesic MCI is related more to vascular disease than diabetes [25], whereas amnesic MCI is considered a phase between normal cognitive function and dementia [26]. The last stage involves serious variations in many cognitive domains with severe consequences in daily life and self-management [27]. 

The association between T2DM and cognitive decline has been confirmed by brain images, indicating reduced brain volume in individuals with T2DM [28]. Loss of white matter is most evident in the frontal and temporal regions, whereas loss of gray matter is most evident in the medial temporal, anterior cingulate, and medial frontal lobes [29]. Regarding the risk of MCI in T2DM patients, epidemiological studies have revealed hazard ratios (HRs) of 1.5 and 1.2 for amnesic MCI and non-amnesic MCI [13,25,30,31,32], respectively. In addition, a meta-analysis has indicated a relative risk of conversion of MCI to dementia of 1.7 (1.1–2.4) for individuals with T2DM compared to individuals without diabetes [33]. According to a previous meta-analysis that included 28 observational studies, individuals with diabetes have an increased risk of 73% for all types of dementia, 56% for AD, and 127% for vascular dementia (VaD) [34]. VaD is a very common form of dementia, the second to AD [35], itis a heterogeneous entity, characterized by vascular damage, cerebral infarcts, amyloid angiopathy, due to ischemic or hemorrhagic brain tissue injury [36]. Moreover, VaD includes four phenotypes, based on the underlying physiopathological mechanism and the brain areas involved, and are: subcortical ischemic vascular dementia, poststroke dementia, multi-infarct dementia and mixed dementia [37]. Lifestyle risk factors, such as tobacco, T2DM, obesity are associated with an increased prevalence of dementia both nonvascular and vascular [38]. But when stratified by sex, women with diabetes have a 19% higher risk to be affected by vascular dementia than men [39].In support of this evidence, several hypotheses can be considered. For example a sex disparity in the management and treatment of diabetes, but also biological differences of gender. In fact, Cole et al. (2007) showed that higher levels of endogenous estradiol in postmenopausal women with T2DM increased risk of dementia [40]. Autopsy studies have shown that the stronger diabetes-related risk of VaD found in women is linked to microvascular damage and neuroinflammation [41]. Moreover, prediabetes increases the risk of dementia [42], suggesting that a risk of dementia is not enhanced by diabetes but by other diabetes-associated factors, such as vascular risk factors. Some population-based cohort studies with older patients (average age > 65 years) have adjusted for vascular risk factors, thereby confirming the association between diabetes and dementia [10]. Diabetes mellitus is a complex metabolic disorder that is closely associated with other risk factors for accelerated cognitive decline. Confounding factor for cognitive decline and dementia can be hypertension, atherosclerotic vascular disease, demographic and socioeconomic factors and genetic factors. These factors could increase the risk of dementia in people with diabetes. Several studies analyzed and confirmed the confounding effects of vascular disease and hypertension. Only two studies reported that adjustment of the relative risk of dementia in diabetic subjects for hypertension and other vascular risk factors appeared to have no effect. Most of the studies adjusted the relative risk of dementia in diabetic patients for age and sex. Only one study adjusted the risk of dementia for the APOE genotype. None of the studies adjusted the relative risk of dementia in diabetic patients for ethnic origin. The potential confounding diabetes-related factors, such as pathology duration, glycemic control, microvascular complications and diabetes treatment, were also not evaluated. Only one study examined the relation between diabetes duration and the risk of dementia, and no effect was observed [10].

## 3. Diabetes of the Brain: AD 

The most studied form of dementia is AD, which is a chronic age-related neurodegenerative disease characterized by progressive memory deficit and neuronal loss as well as by changes in behavior and personality that interfere with daily life [43]. According to the WHO, approximately 50% of the population aged 85 years and older are affected by AD, representing 50–75% of the total cases of dementia [44]. Due to aging trends, the incidence of AD is expected to grow to become the primary disease in the following decades [45], and the prevalence of AD in Europe is expected to double by 2050 [46]. According to the Centers for Disease Control and Prevention (CDC), AD is the sixth leading cause of death in industrialized countries [47]. Because the onset of AD is subtle, the diagnosis is usually made 20 years after the first symptoms appear [48]. To date, none of the drugs (rivastigmine, galantamine, acetylcholinesterase inhibitors, donepezil, memantine, and memantine combined with donepezil) approved by the Food and Drug Administration (FDA) for AD is capable of treating the disease, and the goal of these drugs is only to slow the progression of the neurodegenerative process, improving the quality of life [48]. 

AD has an irreversible progression, but differences exist among individuals [49]. Depending on the age of onset, the survival rate of AD ranges from 3 to 9 years [50]. AD can be sporadic with late onset and unknown underlying etiology, while approximately 5% of cases are familial, with half of these cases having an early onset and are typically related to specific genetic mutations in three major genes [i.e., amyloid precursor protein (APP), presenilin 1 (PS1), and presenilin 2] that are transmitted through families [51]. In the sporadic form, aging plays a primary role [48], but many factors are involved in the onset of the disease, such as vascular disease, hypertension, obesity, dyslipidemia, smoking, female sex, Caucasian ethnicity, genetic factors, history of head trauma, and familial history of dementia [52]. One of the major genetic factors that influences the risk of developing AD at a later age has been identified in the apolipoprotein E (APOE) gene. A variant of the APOE gene, namely, APOE-4, confers an increased risk of developing AD, and it has been estimated that 25% of the population are carriers of APOE4 [53]. In addition, modifiable risk factors account for approximately 40% of the worldwide risk of any type of dementia [54]. Distinctive pathological features include extracellular amyloid plaques and intracellular neurofibrillary tangles (NFTs), resulting in progressive neuronal loss [43] and neurodegeneration, especially in the hippocampus and olfactory epithelium [55].

Some studies have suggested that AD is a metabolic disease strongly related to T2DM [43,56,57,58] and that T2DM is known to promote vascular lesions and ischemic changes in the brain, which may promote the onset and progression of AD [59] (Table 1).

The Rotterdam study was one of the first studies to describe a doubled risk of dementia in T2DM patients [32]. Moreover, Mushtaq et al. (2015) found that T2DM patients have a 65% predisposition to develop AD compared to patients without diabetes [60], and other studies have shown an association between T2DM and an increased risk of dementia [7,52,56,61,62]. T2DM has also been associated with an increased risk of developing amnestic MCI, [25] and higher HbA1C levels correlate with lower cognitive function in neuropsychological testing [63]. Several shared pathological characteristics of AD and T2DM have been described, including insulin resistance, inflammation, lipid abnormalities, oxidative stress, mitochondrial dysfunction, glycated end-products, and autophagy [52,60,64,65]. Therefore, AD has been considered a brain metabolic disease, and the concept of type 3 diabetes mellitus has been proposed [66]. Type 3 diabetes is the inability of brain cells to respond to insulin, leading to impaired synaptic function and metabolism [66].

It is now known that insulin and insulin-like growth factor-1 (IGF-1) have an important role in cognitive ability, neural function, neurogenesis control, and synaptogenesis control [67,68]. Insulin is involved in metabolic activities, cellular survival, and synaptic plasticity [69,70]. Moreover, the insulin receptor is expressed in the hippocampus, hypothalamus, amygdala, cortex, olfactory bulb, and entorhinal cortex [71]. In mammals, the hippocampus and olfactory epithelium are involved in insulin synthesis and secretion [72]. In the brain, stimulation of insulin has a neuroprotective action, preventing the formation of toxic beta-amyloid (Aβ), which is the main component of amyloid plaques, a hallmark of AD [73]; conversely, insulin resistance leads to a detrimental effect on cognitive abilities, and it is associated with cognitive decline [74].

Several studies have reported a consistent reduction in insulin, IGF-1, and their receptors in postmortem brain tissue of AD patients, suggesting that insulin and IGF-1 resistance have an important pathophysiological role in the brain of patients with AD [75,76]. Brain insulin resistance and the associated hyperinsulinemia result in Aβ accumulation [77] via several mechanisms; they increase the activity of β-site amyloid precursor protein cleaving enzyme 1 (BACE-1) [77], which is responsible for the proteolysis of APP into a neurotoxic form [78], and they reduce the activity of insulin-degrading enzyme (IDE), which is involved in the degradation of Aβ and the intracellular domain of APP [79]. It has been reported that BACE-1 knockout mice show a 50% decrease in Aβ degradation [80]. In patients affected by familial AD, IDE expression is reduced in some areas of the brain [81]. Moreover, Aβ accumulation stimulates serine phosphorylation of insulin receptor substrate-1 (IRS-1), consequently impairing insulin signaling, releasing inflammatory molecules, and contributing to brain insulin resistance [82]. Brain insulin resistance affects both Aβ and tau metabolism, and it promotes the hyperphosphorylation of the tau protein, which aggregates and forms paired helical fibers (PHFs), further forming NFTs, which are additional pathophysiological features of AD [83]. NFTs have also been observed in the pancreata of transgenic mice with T2DM and AD [84].

A previous study suggested that not only insulin resistance but also oxidative stress and neuroinflammation promote Aβ and NFT accumulation [85]. Oxidative stress is caused by the overproduction of reactive oxygen species (ROS). The hyperglycemia present in individuals with T2DM promotes the production of ROS in the brain by increasing mitochondrial respiration in pericytes and astrocytes. The overproduction of ROS leads to oxidative stress and neuronal loss [86], causing CI. Neuronal loss in AD is thought to occur primarily via apoptosis [87]. ROS also activate an inflammatory cascade by upregulating nuclear factor kappa B (NF-κB), activator protein-1 (AP-1), and signal transducer and activator of transcription (STAT) pathways [88], resulting in the overproduction of inflammatory cytokines [89].

Neuroinflammation is due to the release of inflammatory cytokines, such as tumor necrosis factor (TNF)-α, interleukin (IL)-6, IL-12, and IL-1β, which is promoted by oxidative stress, insulin resistance [90], and Aβ-induced activation of microglial cells [91]. The release of these cytokines leads to the activation of stress kinases, which phosphorylate the IRS-1 serine residue [92]. Consequently, a vicious pathological cycle of insulin dysregulation, neuroinflammation, and oxidative stress is triggered [93], leading to the neurodegenerative process. Therefore, insulin resistance plays a central role in the association between metabolic disorders and neurodegenerative disorders. Peripheral insulin resistance in T2DM leads to neuroinflammation and impairment of central insulin and IGF-1 signaling as well as metabolic dysfunction, mitochondrial dysfunction, and miRNA deregulation [94], thereby driving neurodegeneration and establishing a strong pathophysiological link between T2DM and AD [82,95] (Figure 1).

Based on the above findings, pharmacological agents approved for T2DM are currently being studied to evaluate their potential usefulness in the treatment of AD.

## 4. GLP-1 Reduces Neuroinflammation

Among the various antidiabetic drugs, glucagon like peptide-1 receptor agonists (GLP-1RAs) play an interesting role in the modulation of neuroinflammation. In addition to its well-known peripheral role, GLP-1 acts as a neurotransmitter and neuromodulator activating central GLP-1 receptors (GLP-1Rs) located in the nucleus of the solitary tract (NTS) and produced in preproglucagon (PPG) neurons [96,97]. GLP-1 plays neurotropic and neuroprotective role in central nervous system [98], preclinical studies have shown that GLP-1Rs exert anti-inflammatory effect in the brain [99] in addition, GLP-1R agonists reduce Aβ aggregation/deposition and hyperphosphorylation of tau protein, oxidative stress, neuronal apoptosis, increasing cell proliferation and neurogenesis [100]. In rodents, GLP-1Rs are also expressed in some hippocampal regions, and GLP-1 improves synaptic plasticity, cognition, and cell survival [101]. There have been several in vitro and in vivo studies on the neuroprotective role of GLP-1RAs in experimental models of AD but few in humans, in the following paragraphs the main studies will be discussed.

## 5. In Vitro and Animal Evidence

In one of the first studies, McClean et al. (2010) demonstrated that liraglutide influences brain neurotransmission in rats and affects synaptic plasticity through enhancing long-term potentiation [102]. Subsequent studies have shown that pretreatment with bilateral intrahippocampal injection of liraglutide improves learning and memory deficit in murine models of AD. Additionally, McClean et al. (2014) investigated the role of liraglutide in APP/PS1 mice, a model of AD, and they reported that liraglutide improves cognitive function, decreases inflammation, decreases amyloid plaque deposition, and increases synapse numbers in the hippocampus [103]. McClean et al. (2011) also demonstrated that intraperitoneal injection of liraglutide in APP/PS1 mice prevents synapse loss in the hippocampus and reduces Aβ plaques in the cortex by 40–45%. Moreover, liraglutide treatment of mice reduces the inflammation measured by microglial activation [104]. Long-Smith et al. (2011) demonstrated that intraperitoneal injection of liraglutide for 8 weeks in mice decreases insulin resistance, microglial activation, and amyloid plaques [105]. In an AD mouse model, another group reported that chronic treatment with liraglutide ameliorates neurogenesis by increasing mature neuron cellular differentiation [106]. Yang et al. (2013) reported that liraglutide decreases abnormal tau protein phosphorylation in T2DM rats [107].

Palleria et al. (2017) evaluated the effects of liraglutide (300 μg/kg daily subcutaneously for 6 weeks) on hippocampal neurodegeneration in a streptozotocin (STZ)-induced diabetes rat model. Although liraglutide improves learning and memory in STZ-treated animals, it also has anxiolytic and pro-depressive effects. In addition, the neuroprotective effects of liraglutide occur via the mammalian target of rapamycin (mTOR) pathway [108]. In a recent study using both in vitro and in vivo AD models, researchers demonstrated a protective role of liraglutide by investigating tau activation and BACE-1 expression using the SH-SY5Y human neuroblastoma cell line. BACE-1 is a transmembrane aspartate protease that cleaves the β-site of APP. Pretreatment of SH-SY5Y cells with liraglutide followed by treatment with okadaic acid (OA) induces apoptosis and increases tau activation [87]. Flow cytometry analysis of neuronal apoptosis and western blot analysis of BACE-1 expression in AD rats demonstrated that liraglutide reduces OA-induced apoptosis and decreases tau activation. Another study in SH-SY5Y human neuroblastoma cells demonstrated that GLP-1 treatment stimulates neuronal viability due to antiapoptotic, antioxidative, and neurotrophic mechanisms, such as increased protein kinase A (PKA) and phosphoinositide 3-kinase (PI3K) pathway activity as well as increased expression of antiapoptotic factors and reduced expression of apoptotic factors [109].

Using an AD mouse model, Park et al. (2021) showed that neuronal apoptosis is mediated by microglia-astrocyte activation via proinflammatory molecules, and they reported that NLY01, an engineered exedin-4, blocks microglial activation of reactive astrocytes, which improves memory and survival of neurons [110]. In a model of neuroinflammation induced by lipopolysaccharide, IL-1β, and hydrogen peroxide (H2O2), Iwai et al. (2014) demonstrated that treatment with a GLP-1 (7–36) amide improves memory and synaptic plasticity in the hippocampus. In addition, it has been reported that lixisenatide plays an important role in neuroprotection [111]. In an APP/PS1 AD model, mice treated with lixisenatide (10 nmol/kg intraperitoneal once daily) for 60 days, exhibited reduced amyloid plaques and neuroinflammation in the hippocampus [112].

Recently, Zhang et al. (2022) found that 5 × FAD transgenic mice treated with the synthetic GLP-1 receptor agonist exenatide had enhanced cognitive function. Moreover, they showed reduced neuroinflammation in extracted piriform cortexes of exenatide-treated mice as well as lower oxidative stress and inflammation in astrocytes. Together, their results suggest that GLP-1 analogs improve cognitive dysfunction in vivo and protect astrocytes in vitro, hypothetically via the downregulation of the NLRP2 inflammasome [113].

Similarly, Qianet al. (2022) demonstrated that GLP-1R activation attenuated microglia-induced neuroinflammation in vitro and in vivo. In addition, activation of GLP-1R in microglia inhibited production of reactive astrocytes. They also uncovered less neuroinflammation, reactive astrocytes, improved myelin integrity, enhanced histology, and amended locomotor function in mice treated with Ex-4 by mediation of the PI3K/ARAP3/RhoA signaling pathway [114].

Interestingly, it has been reported that liraglutide ameliorates brain insulin resistance by inhibiting the c-Jun N-terminal kinase (JNK) pathway and activation of the B-cell lymphoma 2 (Bcl2) gene in T2DM mice [115]. Moreover, GLP-1 stimulates neurogenesis from neuronal progenitor stem cells, and GLP-1 analogues also promote neurogenesis in the hippocampus of AD model mice via mitogen activated protein kinases (MAPKs) [116,117,118].

In a recent study conducted using a murine hippocampus model, Zhang et al. (2021) demonstrated that exenatide acetate (Ex-4), which activates GLP-1R in mice with neuropathic pain, ameliorates memory deficit via several pathways. Ex-4 administration decreases inflammatory signaling by inhibiting the phosphorylation of NF-kB, decreasing the expression of several cytokines (IL-1β and TNF-α), and upregulating postsynaptic density protein 95 (PSD95), which participates in synaptic plasticity [119,120].

## 6. Clinical Evidence

Although there is little clinical evidence showing a role of GLP-1 in cognitive function, promising in vivo and in vitro results have prompted the scientific community to perform clinical trials aimed at testing the neuroprotective role of GLP-1RAs in humans. In 2010, researchers initiated a pilot double-blinded, randomized phase 2 clinical trial with exendin-4 in 57 early-onset AD patients, testing the safety and effects of exendin-4 compared to placebo. The researchers monitored the following outcomes in patients at least 60 years of age with early-stage AD or MCI: clinical progression of dementia [Clinical Dementia Rating (CDR) sum-of-boxes (SB) score]; cognitive performance; brain magnetic resonance imaging (MRI); and various chemicals measured in the blood and cerebrospinal fluid (such as CSF A42, tau, p181-tau, and plasma A42/A40). The researchers analyzed the data of 27 participants, but the pilot study was terminated early at 18 months due to inconclusive results [121]. In a 26-week double-blinded, randomized clinical trial, other researchers enrolled 38 participants with early/moderate AD and tested the effects of liraglutide compared to placebo by evaluating the changes in the intracerebral amyloid deposits in the central nervous system (CNS) in patients with AD assessed by Pittsburgh compound B (PIB) positron emission tomography (PET) scans. Although no changes in amyloid deposition or cognition [122,123] were reported, these negative results may have been due to a short follow-up time. A phase 2 multicenter, randomized, double-blinded study (ELAD study) has enrolled 204 subjects with mild AD treated with liraglutide or matching placebo. The primary outcome is the change in cerebral glucose metabolic rate from baseline to follow-up at 12 months in two groups. The secondary outcomes are as follows: change in clinical and cognitive scores; change in magnetic resonance imaging volume; reduction in microglial activation and in tau formation; and change in amyloid levels and others points. The ELAD study is currently closed, and anticipated results suggest that liraglutide induces changes in the brain but does not improve cognitive measures. However, the official results have not yet been published [124].

Wu et al. (2018) enrolled 106 patients with T2DM and evaluated the association between serum GLP-1 concentration and mild cognitive function impairment in MCI, and they reported that lower serum GLP-1 levels are closely correlated to cognitive dysfunction in patients with diabetes [125]. Valdini et al. (2020) enrolled 40 participants with prediabetic conditions or with new onset T2DM, and the subjects received liraglutide (1.8 mg daily) or lifestyle counseling; they reported that liraglutide slows down cognitive decline as demonstrated by a psychological test evaluating attention, memory, and executive control [126]. Conversely, some negative reports on liraglutide have been reported. In a previous study, 20 T2DM patients were treated with liraglutide (0.6 to 1.8 mg daily for 6 months), and no significant effects on amyloid plaque levels were observed [122]. In another study, 20 obese patients with schizophrenia were treated with exenatide (once weekly for 3 months), and no significant effects on cognitive functions were observed [127].

Another study evalueted amelioration of cognitive decline in patients with T2DM during treatment with liraglutide and its correlation with metabolic improvement. This study is a prospective, parallel-group, open- label, phase III study. The partecipants were inadequately controlled with oral antidiabetic drugs or insulin, they were randomized to receive liraglutide (*n* = 24) or placebo + standard of care group (*n* = 23) for 12 weeks. All patients underwent brain activation monitoring by functional near-infrared spectroscopy and neuropsychological cognition tests to evaluate memory, executive function, attention and verbal function. At the end of study, the liraglutide group demonstrated better scores in all cognitive tests, above all as regard memory and attention (*p* = 0.04) when compared to control group. Moreover, liraglutide increased activation of the dorsolateral prefrontal cortex and orbitofrontal cortex brainregions (*p* = 0.0038) and the improvement of cognitive decline in this group was correlated with morphological changes in brain regions (*p* < 0.05), compared to metabolic changes [128].

Recently, Cheng et al. (2022) carried out a randomized, prospective, open-label, parallel-group trial to investigated the comparative effects of liraglutide, dapagliflozin and acarbose in patients with T2DM. The subjects were inadequately controlled with metformin, so were randomized to receive for 16 weeks liraglutide, dapagliflozin or acarbose. At baseline and post treatment, all participants underwent a brain functional MRI scan and a battery of cognitive assessments (such as Mini-Mental State Examination, Montreal Cognitive Assessment, MoCA). This study involved 36 patients, randomized 1:1:1 (liraglutide, *n* = 12; dapagliflozin, *n* = 12; acarbose, *n* = 12). In the group treated with liraglutide, a significant improvement in impaired odor-induced left hippocampal activation and an improvement of cognitive subdomains of delayed memory, attention and executive function were demonstrated (all *p* < 0.05), unlike the groups treated with dapagliflozin or acarbose [129].

A post-hoc analysis of a phase 3 randomized, double-blinded, placebo-controlled trial (REWIND) evaluated the effect of dulaglutide on CI in T2DM. This study involved 9901 participants, who were randomized to receive dulaglutide (*n* = 4949) or placebo (*n* = 4952). During the follow-up of 5.4 years, cognitive function was assessed in a subset of 8828 subjects (4456 received dulaglutide and 4372 received placebo) using the MoCA and Digit Symbol Substitution Test (DSST). CI occurred in 4·05 per 100 patient-years in dulaglutide group and 4·35 per 100 patient-years in placebo one, but after post-hoc adjustment, the harzard of CI in people with T2DM aged 50 years or above who had further cardiovascular risk factors was reduced by 14% of subjects in the dulaglutide arm (HR 0.86, 95% confidence interval 0.79–0.95; *p* = 0.0018). This analysis showed that long-term treatment with dulaglutide may decrease CI in patients with T2DM [130]. 

Recently, Anita et al. (2021) conducted a meta-analysis of 44 studies that analyzed blood inflammatory markers among T2DM patients with and without CI to evaluate concentration differences of these markers and whether the data were in agreement among studies. They reported that the blood concentrations of IL-6, C-reactive protein (CRP), soluble vascular cell adhesion molecule-1 (sVCAM-1), and advanced glycation end products (AGEs) were significantly higher in CI patients than in those without CI. Brain derived neurotropic factor (BDNF) concentrations were instead significantly lower Adiponectin, C-peptide, homocysteine, intercellular adhesion molecule-1 (ICAM-1), IL-1, leptin, or TNF-α concentrations were not significantly different between two groups. These results demonstrated that CI in T2DM may be promoted by a vascular/inflammatory pattern, but the considerable variance in effect sizes among studies (evaluated by the meta-analysis) indicate that other factors may be implicated in the genesis of CI in T2DM patients [131].

The outcomes and the results of the discussed studies are summarized in Table 2.

## 7. Future Directions

In addition to GLP-1RAs, dual incretin agonists that stimulate both GLP-1R and GIP receptor (GIPR) have been found to reduce markers of neuroinflammation and neurodegeneration [132]. Holscher et al. developed several GLP-1R/GIPR dual agonists, such as DA-JC4 and DA-JC1. In pre-clinical studies, these dual agonists have been demonstrated to reduce synapse loss, DNA damage, proinflammatory cytokines, phosphorylated tau levels, and Aβ plaque deposition as well as to improve memory failure and neurogenesis [133,134,135]. Moreover, the neurotrophic activity of the dual agonist has been reported to be significantly higher than equimolar concentrations of single GLP-1R or GIPR agonists [136]. Additionally, twincretins induce other neuroprotective benefits, such as reduction of oxidative stress and improvement of synaptic health in mice [137].

Triple agonists are peptides that simultaneously stimulate GLP-1R, GIPR, and glucagon receptor (GcgR). In addition to metabolic effects, triple agonists reduce neuroinflammation and neurodegeneration. The triple agonists have the following effects in mouse models: improve cognition, working memory, and long-term spatial memory [138]; reduce Aβ plaques, neuroinflammation, and oxidative stress; and stimulate neurogenesis and synapse number [139]. Similar to dual agonists, triagonists also increase cyclic adenosine monophosphate (cAMP) levels higher than a single GLP-1R agonist on an equimolar basis [140]. Moreover, preclinical studies have indicated that GLP-1/GIP/GcgR agonists improve memory behavior, synaptic transmission, neuronal excitability, and Ca^2+^ homeostasis in 3xTg-AD mice [141]. Similarly, triagonists induce neurotrophic and neuroprotective actions, mitigating oxidative stress and glutamate excitotoxicity in mouse models of mild traumatic brain injury. Therefore, these data suggest that multireceptor agonists may be a novel therapeutic strategy for the treatment of both cognitive dysfunction and AD.

## 8. Conclusions

Cognitive disorder is an important issue in global health that impairs the quality of life. It is known that T2DM predicts several neuronal injuries, and the risk of dementia is rapidly increasing. To date, the available neurological treatments are symptomatic but do not influence the course of the disease.

In recent decades, researchers have focused on the potential effect of antidiabetic drugs on the brain. The effectiveness of antidiabetic drugs, such as metformin and sulfonylureas, on the development of CI has not been demonstrated, and the potential hypoglycemia induced by sulfonylureas worsen CI. Moreover, preclinical data have indicated that thiazolidinediones and gliptins do not have beneficial effects, but gliptins, which increase GLP-1 levels in the brain, improve cognitive decline in individuals with diabetes [142]. Thus, GLP-1 agonists may influence cognitive decline in individuals with diabetes. Preclinical models have demonstrated that treatment with liraglutide and exenatide reduces neuroinflammation, tau phosphorylation, and amyloid deposition, thereby improving cognitive outcomes and facilitating the development of neuronal progenitor cells. The association between GLP-1RA and ameliorating cognitive function in T2DM patients is preliminary but interesting. These treatments may modify the course of MCI and dementia. Further, GLP-1RA may be a promising therapeutic treatment to improve neuronal impairment by modifying several pathophysiological pathways, and additional studies will be conducted in the future.

## Figures and Tables

**Figure 1 ijms-24-11301-f001:**
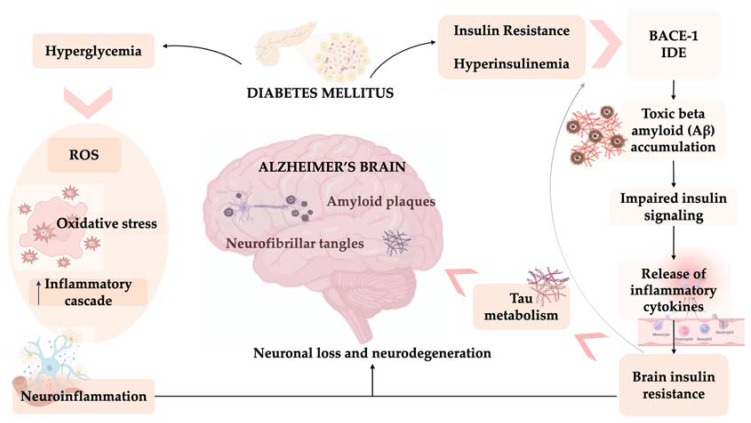
Neuroinflammation and diabetes of the brain. Alzheimer’s disease (AD) is characterized by extracellular amyloid plaques and intracellular neurofibrillary tangles (NFTs), resulting in progressive neuronal loss and neurodegeneration, especially in the hippocampus and olfactory epithelium. The correlation between AD and type 2 diabetes mellitus (T2DM) includes insulin resistance, inflammation, and oxidative stress. Brain insulin resistance and the associated hyperinsulinemia increase the activity of β-site amyloid precursor protein cleaving enzyme 1 (BACE-1), which is responsible for the proteolysis of amyloid precursor protein (APP), resulting in a neurotoxic form. Furthermore, brain insulin resistance and the associated hyperinsulinemia reduce the activity of insulin degrading enzyme (IDE), which is involved in the degradation of β-amyloid (Aβ) and the intracellular domain of APP. Accumulation of Aβ, in turn, impairs insulin signaling and releases inflammatory molecules, thereby contributing to brain insulin resistance. Brain insulin resistance affects both Aβ and tau metabolism; it promotes tau protein hyperphosphorylation, which forms paired helical fibers (PHFs) and NFTs. Hyperglycemia promotes the production of reactive oxygen species (ROS), leading to oxidative stress; moreover, ROS activate an inflammatory cascade by upregulating the nuclear factor kappa B (NF-κB), activator protein-1 (AP-1), and signal transducer and activator of transcription (STAT) pathways, consequently inducing the overproduction of inflammatory cytokines. Neuroinflammation is due to the release of inflammatory cytokines, such as tumor necrosis factor (TNF)-α, interleukin (IL)-6, IL-12, and IL-1β, which are promoted by oxidative stress and insulin resistance. Consequently, a pathological vicious cycle involving insulin dysregulation, neuroinflammation, and oxidative stress is triggered, leading to the neurodegenerative process.

**Table 1 ijms-24-11301-t001:** Studies explaining potential therapeutic strategy of AD ([53,54,55,59]).

Study	Aim of the Study	Results
Michailidis M. et al. (2022) [53]	Evaluate the efficacy of antidiabetic drugs in AD treatment	Intranasal insulin improves: cognitive, verbal, working and audiovisual memory and functional ability. Metformin: inconclusive results because some studies demonstrated that it reduces risk of CI and Dementia and improves verbal memory, attention and executive function; other studies demonstrated that its long-term use increases risk of CI and Dementia. Liraglutide: improves cerebral glucose metabolism. So it has moderate neuroprotective effects. Rosiglitazone: improves attention and delayed recall and protects against cognitive impairment. Pioglitazone: improves cognition, metabolism and cerebral blood flow to the parietal lobe.
Livingston G. et al. (2020) [54]	Evaluate prevention strategy and treatment strategy	Modifying risk factors (such as hypertension, tobacco and alcohol use, obesity, noise exposure etc.) might prevent or delay up to 40% of dementias. For patients with dementia: it’s necessary farmacological help (to reduce neuropsychiatric symptoms) and social support for patient and family carers.
Yao Z. G. et al. (2016) [55]	Analyse the effects of Valproic Acid on olfactory dysfunction of APP/PS1 double transgenic mouse models of AD	Valproic Acid improves olfactory performances, reduces β-amyloid deposition in olfactory epithelium, decreases cell apoptosis of olfactory epithelium, reduces senile plaques and levels of soluble and insoluble Aβ42 peptides in olfactory bulb so it improves olfactory performances and prevented degenerative changes.
Ohyagi Y. et al. (2019) [59]	Analyse of apomorphine that promoted intracellular Aβ degradation and improved memory function	Apomorphine treatment reduces neuronal insulin resistance and activates insulin-degrading enzyme, an Aβ-degrading enzyme.

**Table 2 ijms-24-11301-t002:** Studies comparing effects of GLP1-receptor agonist on risk of dementia.

Study	Population	Sample Size	Intervention	Outcome	Results
Mullins R.J. et al. (2019) [121]	Patients (more than 60 years old) with evidence of early-stage Alzheimer’s disease or mild cognitive impairment in screening testing.	27 Exenatide (*n* = 13) vs. placebo (*n* = 14)	Exenatide 5 mcg or 10 mcg SC twice daily vs. Placebo	Safety and tolerability of exenatide and responses for clinical, cognitive, and biomarker outcomes in early Alzheimer Disease.	Exenatide treatment produced no differences on Alzheimer Disease outcome.
Egefjord L. et al. (2012) [122]	Patients from 50 years to 80 years with Alzheimer Disease (AD) diagnosis.	34 Liraglutide vs. Placebo	Liraglutide 1.8 mg vs. Placebo	Liraglutide will change the intra-cerebral amyloid deposit in the CNS in patients with Alzheimer’s disease assessed by PIB PET scan.	Closed, but no results posted.
Gejl M. et al. (2016) [123]	Patients from 50 years to 80 years with Alzheimer Disease (AD) diagnosis.	38 Liraglutide (*n* = 18) vs. Placebo (*n* = 20)	Liraglutide 1.8 mg daily vs. Placebo	Change in deposition ofamyloid deposit (Aβ) in the Central Nervous System assessed by PIB PET scan followed by improvement of cognition.	In Liraglutide group, the Glucose metabolism (CMR_glc_)increased numerically but non-significantly (all *p* ≥ 0.49) compared to Placebo group in which Glucose metabolism (CMR_glc_) declined significantly (*p* = 0.04).
Femminella G.D. et al. 2019 [124]	Patients (more than 50 years old) with mild Alzheimer’s dementia.	206 Liraglutide (*n* = 103) vs. Placebo (*n* = 103)	Liraglutide 1.8 mg vs. Placebo	Change in cerebral glucose metabolic rate in the cortical regions from baseline to follow up (12 months).	Closed, but no results posted.
Vadini F. et al. (2020) [126]	Patients with IFG or T2DM, BMI > 30 in treatment with diet therapy plus metformin at the highest tolerated dose.	32 Liraglutide (*n* = 16) vs. Placebo (*n* = 16)	Liraglutide 1.8 mg vs. Placebo	Liraglutide effects on cognitive functions in type 2 diabetic subjects independently on the weight loss it might induce.	Liraglutide significantly increased short term memory (*p* = 0.024) and memory composite z-score (*p* = 0.0065).
Li et al (2021) [128]	Patients aged 18 to 65 years with a glycated hemoglobin (HbA1c) value of >7.0%, whowere treated with oral antidiabetic drugs or insulin for at least 3 months.	47 Liraglutide (*n* = 24) Vs Placebo (*n* = 23)	Liraglutide from 0.6 mg to 1.8 mg daily vs. placebo	Change in scores of neuropsychological cognition tests and brain activation monitoring by functional near-infrared spectroscopy	At 12 weeks, the liraglutide group demonstrated better scores in all cognitive tests, above all as regard memory and attention (*p* = 0.04) when compared to control group. Liraglutide increased activation of the dorsolateral prefrontal cortex and orbitofrontal cortex brain regions (*p* = 0.0038) and the improvement of cognitive decline in this group was correlated with morphological changes in brain regions (*p* < 0.05), compared to metabolic changes.
Cheng et al. (2022) [129]	All patients were aged 40 to 75 years, had hemoglobin A1c (HbA1c) of 7.0–10.0% and a BMI of $25 kg/m^2^, and had been on a stable dose of metformin monotherapy (1500 mg daily) for at least 90 days.	36 Liraglutide (*n* = 12); dapagliflozin (*n* = 12); acarbose (*n* = 12)	Liraglutide from 0.6 mg to 1.8 mg once daily; dapagliflozin 10 mg once daily; acarbose from 50 mg to 100 mg threetimes daily	The primary end point was the change in odor-induced brain activation from baseline to week 16 evaluated with fMRI and cognitive assessments.	In the group treated with liraglutide, a significant improvement in impaired odor-induced left hippocampal activation and an improvement of cognitive subdomains of delayed memory, attention and executive function were demonstrated (all *p* < 0.05), unlike the groups treated with dapagliflozin or acarbose.
Cukierman-Yaffe T. et al. (2020) [130]	Patients more than 50 years old with T2DM and additional cardiovascular risk factors, glycated haemoglobin of up to 9 5% on a maximum of two oral glucose-lowering drugs with or without basal insulin and a body-mass index of at least 23 kg/m^2^.	8828 Dulaglutide (*n* = 4456) vs. Placebo (*n* = 4372)	Dulaglutide 1.5 mg once weekly vs. Placebo	Exploratory analysis within REWIND trial to evaluate association between dulaglutide and cognitive impairment.	Long-term treatment with dulaglutide might reduce cognitive impairment that occours in 4 05 per 100 patient-years in participants assigned dulaglutide and 4 35 per 100 patient-years in people assigned placebo.

## Data Availability

Not applicable.

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
