# Peer review of "Potential Role of Glucagon-like Peptide-1 Receptor Agonists in the Treatment of Cognitive Decline and Dementia in Diabetes Mellitus"

_ijms, 2023, doi:10.3390/ijms241411301_

Round 1
Reviewer 1 Report
In this manuscript, Potential role of glucagon-like peptide-1 receptor agonists in the treatment of cognitive decline and dementia in diabetes mellitus was studied well. But there are some questions in the aspects of Introduction, Clinical evidence and Diabetes of the brain: AD and so on.
Hence, I have some suggestions as follows:
1) Some descriptions in the manuscript were not exact or confusing. Some words which will make the manuscript feel like an article on a popular science book should not appear in such a research paper. The following are suggestions for improving English usage. Please use standard expression in English.
2) Please add some more figure analysis.
3) The manuscript stays within a stage of literature survey, and is hard to find original contribution of the authors on this subject.
4) Problems on format or details: the manuscript was not well prepared according to the “Guidelines”. Please check carefully.
5) If you put some photos of your design into the paper, the design will be more clearly understood.
Moderate editing of English language required
Author Response
In this manuscript, Potential role of glucagon-like peptide-1 receptor agonists in the treatment of cognitive decline and dementia in diabetes mellitus was studied well. But there are some questions in the aspects of Introduction, Clinical evidence and Diabetes of the brain: AD and so on.
Hence, I have some suggestions as follows:
We would like to thank the Reviewer for his/her comments.
Q1) Some descriptions in the manuscript were not exact or confusing. Some words which will make the manuscript feel like an article on a popular science book should not appear in such a research paper. The following are suggestions for improving English usage. Please use standard expression in English.
R1)We apologize for the lack of clarity. According also with the constructive suggestions of the Reviewer 2, the manuscript has been extensively revised.
In the submitted version of the manuscript the English language had been revised by Charlesworth Author Services as reported in page 13, line 505.
Q2) Please add some more figure analysis.
R2) The figure has been completely modified
Q3) The manuscript stays within a stage of literature survey, and is hard to find original contribution of the authors on this subject.
R3)Dear Reviewer, we have been invited by Dr. Alessandra Puddu and Dr. Davide Maggi, Guest Editors of the Special Issue on " Anti-Inflammatory Effects of Glucagon-Like Peptide-1 " in the International Journal of Molecular Sciences to submit a review on this issue.
The focus of this Special Issue was “any aspect of pancreatic and extra-pancreatic anti-inflammatory effects of GLP-1. The implication not only for diabetes, but also for other diseases is of particular interest.” We have submitted the title “Potential role of glucagon-like peptide-1 receptor agonists in the treatment of cognitive decline and dementia in diabetesmellitus” The title has been accepted by the Guest Editors and we have then prepared the manuscript according to it and to the Focus of this “Special Issue”
Q 4) Problems on format or details: the manuscript was not well prepared according to the “Guidelines”. Please check carefully.
R4) Done
Please see the attachment

Reviewer 2 Report
Reviewer comments and suggestions
The authors in this study discussed the correlations between diabetes and cognitive decline as well as between diabetes and AD, stressing underlying pathophysiological mechanisms of diabetic cognitive dysfunction and brain insulin resistance. Additionally, they present the data of glucagon-like peptide-1 (GLP-1) on neuroinflammation and its possible therapeutic role in both dementia and AD.The drug was reported to act on synaptic plasticity, cognition, and cell survival.
Overall, the manuscript was well written. However, a few concerns/comments needed to be explained/modified.
- Line 42-43 one reference was not enough
- Line 51 please explain the study
- Line 80-82 probably this was due to ageing there were age related differences in type 1 and type2 diabetes, need to explain it clearly
- Line 101-102 more references for this
- Line 107, please describe vascular dementia
- Line 108 Is there any possible reason for this, reference 22
- Line 111-112 Please explain the study about confounding factor
- Line 126-128 Please discuss the study, reference 30 and 31
- Line 141-148 It would be nice if the authors can tabulate the results.
- Line 174-175 Please add references
- Figure 1 Image should be modified.
- Line 242-244 With the help of the reference cited, the authors could enhance the information regarding this section
- Line 350-351 Please explain the results
- Line 355-359 Please avoid long sentences.
- In line 393 a typo error was present
- The journal style should be modified based on the author’s guidelines of MDPI.
Author Response
The authors in this study discussed the correlations between diabetes and cognitive decline as well as between diabetes and AD, stressing underlying pathophysiological mechanisms of diabetic cognitive dysfunction and brain insulin resistance. Additionally, they present the data of glucagon-like peptide-1 (GLP-1) on neuroinflammation and its possible therapeutic role in both dementia and AD. The drug was reported to act on synaptic plasticity, cognition, and cell survival.
Overall, the manuscript was well written. However, a few concerns/comments needed to be explained/modified.
We would like to thank the Reviewer for his/her comments that have helped us to improve the manuscript. According to his/her suggestions, we have now modified the manuscript as follows.
Q1) Line 42-43 one reference was not enough.
R1) The authors agree. As suggested by Reviewer we have added the following references (4,5 and 6):
- Akter, K.; Lanza, E. A.; Martin, S. A.;Myronyuk, N.; Rua, M.; Raffa, R. B. Diabetes mellitus and Alzheimer's disease: shared pathology and treatment?. Br J Clin Pharmacol 2011, 71(3), 365–376. https://doi.org/10.1111/j.1365-2125.2010.03830.x.
- Baglietto-Vargas, D.; Shi, J.; Yaeger, D. M.; Ager, R.;LaFerla, F. M. Diabetes and Alzheimer's disease crosstalk. Neurosci Biobehav Rev 2016, 64, 272–287. https://doi.org/10.1016/j.neubiorev.2016.03.005.
- Mamelak M. Energy and the Alzheimer brain. Neurosci Biobehav Rev 2017, 75, 297–313. https://doi.org/10.1016/j.neubiorev.2017.02.001.
Q2) Line 51 please explain the study.
R2) Done. We would like to thank the Reviewer for pointing out this aspect. As suggested by Reviewer, the study has been explained and further information from the review cited in reference 7 and new references have been added in the manuscript (Introduction, Page 2, line 57 and following). “Biessels, G.J et al (2006) reported that the incidence of Alzheimer's disease and vascular dementia was higher in subjects with diabetes than in those without diabetes. This paper suggested that impaired glucose metabolism and vascular disease underlies the pathophysiology of dementia. In particular, chronic hyperglycemia in diabetes can lead to brain structure abnormalities, such as thickening of the basement membrane of cerebral capillaries and ultimately chronic brain ischemia. Similarly, hypoglycemia frequently associated with insulin treatment can cause irreversible structural brain abnormalities . Indeed, several studies demonstrated that the risk of dementia was highest in insulin-treated diabetic people . Director indirect effects of insulin could contribute to the risk of dementia. Hypoglycemic episodes frequently complicate insulin treatment and contrary to hyperglycemia, hypoglycemia can cause irreversible brain damage”.
Q3) Line 80-82 probably this was due to ageing there were age related differences in type 1 and type2 diabetes, need to explain it clearly.
R3) We thank the Reviewer for pointing out this important issue. According to this suggestion we have discussed this point and a new sentence has been added in the Section: 2. Diabetes, cognitive dysfunction and dementia (page 3, lines 99 and following) “So there are differences between T1DM and T2DM: the first one appears to be associated to a slow-down of mental processing and executive/attentional functioning, while the second one has additionally memory problems. Hyperglycemia and microvascular complications are risk factors that we found both in T1DM and T2DM. However, T2DM is diagnosed at an older age than T1DM (old age is in itself a risk factor for dementia) and it occurs with obesity, insulin resistance, hypertension, and dyslipidemia, which have negative effects on the brain. On the other hand, T1DM is almost always diagnosed during childhood and adolescence that are periods of development of central nervous system and younger brain is more susceptible to extreme values of glycemia.
Q4) Line 101-102 more references for this.
R4) The authors agree. We have now added the following references (13, 30,31 and 32):
- Ott, A.;Stolk, R. P.; van Harskamp, F.; Pols, H. A.; Hofman, A.; Breteler, M. M. Diabetes mellitus and the risk of dementia: The Rotterdam Study. Neurology 1999, 53(9), 1937–1942. https://doi.org/10.1212/wnl.53.9.1937
- Leibson, C. L.; Rocca, W. A.; Hanson, V. A.; Cha, R.; Kokmen, E.; O'Brien, P. C.; Palumbo, P. J. Risk of dementia among persons with diabetes mellitus: a population-based cohort study. Am J Epidemiol 1997, 145(4), 301–308. https://doi.org/10.1093/oxfordjournals.aje.a009106
- MarszaÅ‚ek M. Cukrzycatypu 2 a choroba Alzheimera - jednaczydwiechoroby? Mechanizmyasocjacji [Diabetes type 2 and Alzheimer disease - one or two diseases? Mechanisms of association]. Postepy Hig Med Dosw (Online) 2013, 67, 653–671. https://doi.org/10.5604/17322693.1059549
- Ott, A.; Stolk, R. P.; Hofman, A.; van Harskamp, F.; Grobbee, D. E.; Breteler, M. M. Association of diabetes mellitus and dementia: the Rotterdam Study. Diabetologia 1996, 39(11), 1392–1397. https://doi.org/10.1007/s001250050588
Q5) Line 107, please describe vascular dementia.
R5) Done. As suggested the vascular dementia has been descrive (Section: 2. Diabetes, cognitive dysfunction, and dementia, Page 3, Line 133 and following).“VaD is a very common form of dementia, the second to AD, it is a heterogeneous entity, characterized by vascular damage, cerebral infarcts, amyloid angiopathy, due to ischemic or hemorrhagic brain tissue injury. Moreover, VaD includes four phenotypes, based on the underlying physiopathological mechanism and the brain are as involved, and are: subcortical ischemic vascular dementia, post stroke dementia, multi-infarct dementia and mixed dementia.”
Q6) Line 108 Is there any possible reason for this, reference 22.
R6) We would like to thank the Reviewer for pointing out this additional aspect. According to this suggestion we have discussed this point and a new sentence has been added in the Section: 2. Diabetes, cognitive dysfunction, and dementia (page 3, lines 142 and following).“In support of this evidence, several hypotheses can be considered. For example a sex disparity in the management and treatment of diabetes, but also biological differences of gender. In fact, Cole et al. (2007) showed that higher levels of endogenous estradiol in postmenopausal women with T2DM increased risk of dementia. Autopsy studies have shown that the stronger diabetes-related risk of VaD found in women is linked to microvascular damage and neuroinflammation.
Q7) Line 111-112 Please explain the study about confounding factors.
R7) We thank the Reviewer for pointing out this important issue. According to this suggestion a new sentence has been added in the Section: 2. Diabetes, cognitive dysfunction, and dementia (page 3, lines 152 and following).
“Diabetes mellitus is a complex metabolic disorder that is closely associated with other risk factors for accelerated cognitive decline. Confounding factor for cognitive decline and dementia can be hypertension, atherosclerotic vascular disease, demographic and socioeconomic factors and genetic factors. These factors could increase the risk of dementia in people with diabetes.Several studies analyzed and confirmed the confounding effects of vascular disease and hypertension. Only two studies reported that adjustment of the relative risk of dementia in diabetic subjects for hypertension and other vascular risk factors appeared to have no effect. Most of the studies adjusted the relative risk of dementia in diabetic patients for age and sex. Only one study adjusted the risk of dementia for the APOE genotype. None of the studies adjusted the relative risk of dementia in diabetic patients for ethnic origin. The potential confounding diabetes-related factors, such as pathology duration, glycemic control, microvascular complications and diabetes treatment, were also not evaluated. Only one study examined the relation between diabetes duration and the risk of dementia, and no effect was observed”.
Q8) Line 126-128 Please discuss the study, reference 30 and 31.
R8) We apologize for the mistake. The references 30 and 31 have been inserted incorrectly in the main text. The correct reference is the following n°48:
- Alzheimer’s association. 2019 Alzheimer’s disease facts and figures. Alzheimers Dement 15: Volume15, Issue 3 March 2019 Pages 321-387 .https://doi.org/10.1016/j.jalz.2019.01.010
Q9) Line 141-148 It would be nice if the authors can tabulate the results.
R9) Done. A new Table (Table1) has been now inserted in the main text.
Q10) Line 174-175 Please add references.
R10) Done. As suggested by Reviewer,wehave added the following references (75 and 76):
- Leclerc, M.; Bourassa, P.; Tremblay, C.; Caron, V.; Sugère, C.; Emond, V.; Bennett, D. A.; Calon, F. Cerebrovascular insulin receptors are defective in Alzheimer's disease. Brain 2023, 146(1), 75–90. https://doi.org/10.1093/brain/awac309;
- Wakabayashi, T.; Yamaguchi, K.; Matsui, K.; Sano, T.; Kubota, T.; Hashimoto, T.; Mano, A.; Yamada, K.; Matsuo, Y.; Kubota, N.; Kadowaki, T.; Iwatsubo, T. Differential effects of diet- and genetically-induced brain insulin resistance on amyloid pathology in a mouse model of Alzheimer's disease. Mol Neurodegener 2019, 14(1), 15. https://doi.org/10.1186/s13024-019-0315-7
Q11) Figure 1 Image should be modified.
R11) According to this suggestion, the Figure 1 has been modified.
Q12) Line 242-244 With the help of the reference cited, the authors could enhance the information regarding this section.
R12) We would like to thank the Reviewer for this suggestion. According to this suggestion a new sentence has been added in the Section: 4. GLP-1 reduces neuroinflammation (page 7, lines 306 and following), GLP-1 plays neurotropic and neuroprotective role in central nervous system, and preclinical studies have shown that GLP-1Rs exert anti-inflammatory effect in the brain. In addition, GLP-1R agonists reduce Aβ aggregation/deposition and hyperphosphorylation of tau protein, oxidative stress, neuronal apoptosis, increasing cell proliferation and neurogenesis.
Q13) Line 350-351 Please explain the results.
R13) Done. As suggested by Reviewer, the results of REWIND Study have been now explained (Section: 6. Clinical evidences, Page 10, line 421 and following) “CI occurred in 4·05 per 100 patient-years in dulaglutide group and 4·35 per 100 patient-years in placebo one, but after post-hoc adjustment, the harzard of CI in people with T2DM aged 50 years or above who had further cardiovascular risk factors was reduced by 14% of subjects in the dulaglutide arm (HR 0.86, 95% confidence interval 0.79–0.95; p=0.0018). This analysis showed that long-term treatment with dulaglutide may decrease CI in patients with T2DM.”
Q14) Line 355-359 Please avoid long sentences.
R14) We apologize with the Reviewer.We reduced the sentence length.
Q15) In line 393 a typo error was present.
R15)Done. We have corrected typo error.
Q16) The journal style should be modified based on the author’s guidelines of MDPI.
R16) The manuscript has been prepared according to the author’s guidelines of MDPI
Please see the attachment

Round 2
Reviewer 1 Report
The paper has been modified a lot, and the current version is basically acceptable
The paper has been modified a lot, and the current version is basically acceptable
Author Response
We wish to thank You for the thoughtful comments that have helped us to improve the manuscript.